DATA RELEASE

# A database of restriction maps to expand the utility of bacterial artificial chromosomes

Eamon Winden[1], Alejandro Vasquez-Echeverri[2], Susana Calle-Castañeda[1], Yumin Lian[1], Juan Pablo Hernandez Ortiz[3] and David C. Schwartz[1,*]

1 Laboratory of Genetics, Department of Chemistry, Laboratory for Molecular and Computational Genomics, University of Wisconsin-Madison, Madison, WI 53705, USA
2 Department of Biophysics, Department of Genetics, Department of Chemistry, Laboratory for Molecular and Computational Genomics, University of Wisconsin-Madison, Madison, WI 53705, USA
3 Departamento de Materiales y Nanotecnología, Universidad Nacional de Colombia – Medellín, Medellín, 050034, Colombia

## ABSTRACT

While Bacterial Artificial Chromosomes libraries were once a key resource for the genomic community, they have been obviated, for sequencing purposes, by long-read technologies. Such libraries may now serve as a valuable resource for manipulating and assembling large genomic constructs. To enhance accessibility and comparison, we have developed a BAC restriction map database. Using information from the National Center for Biotechnology Information's cloneDB FTP site, we constructed a database containing the restriction maps for both uniquely placed and insert-sequenced BACs from 11 libraries covering the recognition sequences of the available restriction enzymes. Along with the database, we generated a set of Python functions to reconstruct the database and more easily access the information within. This data is valuable for researchers simply using BACs, as well as those working with larger sections of the genome in terms of synthetic genes, large-scale editing, and mapping.

**Subjects** Genetics and Genomics, Molecular Genetics, Synthetic Biology

**Submitted:** 03 May 2023

\* Corresponding author. E-mail: dcschwartz@wisc.edu

Preprint submitted at https://doi.org/10.1101/2023.03.31.535162

## DATA DESCRIPTION

### Context

Bacterial Artificial Chromosomes (BACs) were a central resource for early sequencing and mapping efforts that enabled the construction of the first reference genome for humans and other higher eukaryotes [1, 2]. While BACs have been largely supplanted for creating reference genomes [3, 4], research directions in synthetic biology are now advancing towards large-scale genome alterations and assemblies. Consequently, construction technologies are emerging as an increasingly valuable resource for contemporary investigators [5]. Accordingly, BACs could be leveraged as expansive, well-characterized genomic elements to be used as building blocks. In this regard, there are currently limited resources available for fully engaging the many advantages offered by BACs. While there are fully sequenced BACs, including those from the "golden path" used to generate drafts of the human genome, the UCSC Genome Browser presents a collection of end-sequenced BACs [6], and NCBI's cloneDB has been discontinued [4]. Consequently, synthetic efforts would benefit from resources that readily functionalize BACs, transforming them into

physically manipulable molecules, or "parts," for genome writing applications. One such fundamental resource would be a database of restriction maps created from available human BAC clones.

## Data description

BAC DNA molecules require special considerations for synthetic workflows due to their large size and for mediating cloning and sequencing errors – particularly when dealing with complex, repeat-ridden portions of the human genome [7, 8]. Here, a comprehensive set of restriction maps simplifies selecting and validating optimal BACs within libraries for such applications.

## Analysis

We created this resource using publicly available data from the National Center for Biotechnology Information (NCBI) [9]. End-sequenced BACs with unique placement were mapped, as were insert-sequenced BACs. The resulting database of restriction maps includes most of the BACs publicly available for the human genome. As sequencing costs have reduced dramatically, the later addition of more libraries or species would be welcome. The diversity of BAC libraries, assembled from different donors and by different methods, enriches the diversity and utility of the available sequences for uses such as genome writing applications. This resource offers restriction maps available across many BACs, enabling the systematic selection of clones and restriction fragments for a broad range of applications.

## Discussion

This database establishes a useful resource for directed manipulations of large insert clones – BACs. This method is a convenient way to explore sizable, discrete chunks that would readily allow the contextualization of genic regions within non-coding, genomic "dark matter" portions of the genome. With ready access to comprehensive restriction maps, searching for BACs with specific characteristics or developing workflows concerning linearization and vector removal are now enhanced and simplified. Figure 1 shows an output of a simple pipeline determining a clone and fingerprint map from a region of interest and then visually representing it. The UCSC genome browser is an excellent resource and can find a BAC by name and determine its restriction map. However, this browser lacks the utility to create, study and compare an ensemble of these maps, thus inspiring us to create this database.

### RE-USE POTENTIAL

This database is a tool that can provide new insights into the human genome, especially at the scale of hundreds of kilobases. CRISPR-Cas systems have revolutionized genomics. Pairing our database with one of the many guide-RNA libraries to find targets for manipulation with CRISPR tools further synergizes BAC advantages for genomic research. Additionally, our *bacmapping* python package can be expanded for new clones, libraries, and species.

### POTENTIAL IMPLICATIONS

BACs are a versatile resource for well-characterized genomic fragments created for various organisms [11]. While these libraries served well to establish reference genomes, they may



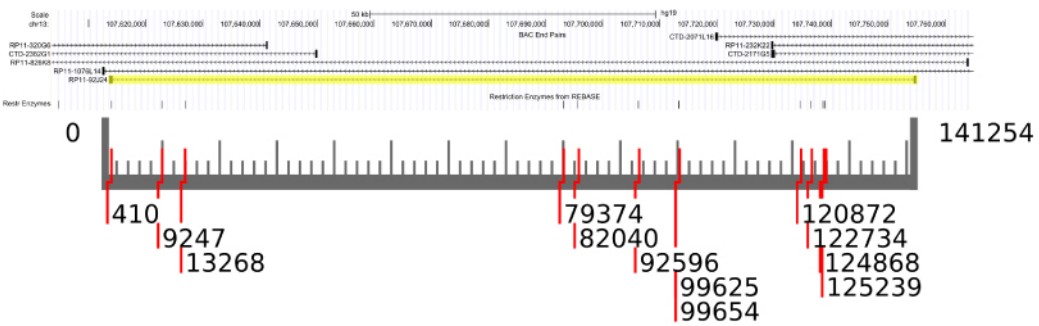

**Figure 1.** (A) A screenshot of the UCSC genome browser [6, 10]. The RP11-92J24 clone is placed in chromosome 13 of the human genome, and the restriction sites for KpnI are shown below. (B) An example of the insert of RP11-92J24 mapped by KpnI and drawn with a Python (RRID:SCR_008394) function, drawMap, which is included in the package bacmapping. The 3′ overhangs are shown to improve clarity.

now be refashioned as a source of large-scale building blocks for genome construction and manipulations. This way, they will empower new ways to investigate chromosome-scale biology [12]. These studies require a resource of BAC components ready to be forged into final products. Accordingly, databases like this one provide the basic information scientists need to start selecting BACs from the available libraries.

## METHODS

Sequence and details regarding each BAC were downloaded from the NCBI's FTP server [9] and curated for quality, focusing on end-sequenced BACs with unique placement of a reasonable length (25–350 kilobases). Individual BACs were processed by a Python pipeline written for distributed processing, and maps were saved for 233 enzymes, representing all available enzyme recognition sites [13]. The pipeline relies on the Bio.Restriction class from Biopython (RRID:SCR_007173) [14], which identifies restriction digest sites for an enzyme in a sequence. Most of the library comprises 281,839 end-sequenced BACs from eight different libraries covering 94% of the genome with an average insert length of 143,476 base pairs. A total of 25,653 insert-sequenced BACs from 37 libraries, with an average insert length of 120,298 base pairs, are also included. A random sample of restriction maps were validated against NEBCutter3 digestions to ensure a functional pipeline [15]. The rare-cutting enzymes were of most interest. To save space, the database was truncated at enzymes that cut a BAC more than 50 times, as such digest would create small fragments easily recreated by PCR. All the scripts used to generate this library are available on the GitHub repository, in case the library should be built locally or updated (RRID:SCR_023940, biotools:bacmapping). Analysis scripts and Jupyter notebooks containing examples for building and using the database are also available to best harness the resource and sequence data related to BACs. These scripts focus on finding maps and sequences for specific BACs, and manipulating these maps to design new experiments.

## AVAILABILITY OF SOURCE CODE AND REQUIREMENTS

Project name: bacmapping
Project home page: https://github.com/Laboratory-of-Mol-and-Comp-Genomics/bacmapping
Operating system(s): Ubuntu 22.04, Windows 11, MacOS Monterey 12.6
Programming language: Python

Other requirements: NumPy (RRID:SCR_008633), pandas 1.5.2 (RRID:SCR_018214), biopython 1.80, MatPlotLib 3.6.3 (RRID:SCR_008624), multiprocessing 0.70.14
License: MIT
RRID:SCR_023940.

## DATA AVAILABILITY

The data presented here is produced by data from cloneDB and is available from GigaDB [16].

## ABBREVIATIONS

BACs, Bacterial Artificial Chromosomes; NCBI, National Center for Biotechnology Information.

## DECLARATIONS

### Ethics approval

The authors declare that ethical approval was not required for this type of research.

### Competing Interests

The authors declare that they have no competing interests.

### Authors' contributions

EW and DCS conceived the original idea. EW and AVE produced the Python programs. AVE, SCC and YL contributed to the framework for the package and concepts. SCC and YL contributed experimental support and testing. JPHO and DCS supervised and contributed computational oversight and guidance in applying the package. EW and DCS wrote most of the manuscript.

### Funding

COLCIENCIAS, Minciencias, Colombia, scholarship program 783; NHGRI R21 HG012281 (DCS).

### Acknowledgements

We thank the good people, past and present, associated with and of the Laboratory of Molecular and Computational Genomics, namely Dr. Sam Krerowicz and Dr. Louise Pape. We also thank NHGRI for funding: R21 HG012281 (DCS). AVE and SCC were partly supported by COLCIENCIAS, Minciencias, Colombia, scholarship program 783.

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
