## [Editor Report]

Editor’s AssessmentWhile Bacterial Artificial Chromosomes libraries were once a key resource for building the human genome project over time they have been rendered relatively obsolete by long-read technologies. In the era of CRISPR-Cas systems pairing this data with one of the many guide-RNA libraries to find targets for manipulation with CRISPR tools is bringing back BACs advantages for genomics. With this in mind the authors have developed a BAC restriction map database containing the restriction maps for both uniquely placed and insert-sequenced BACs from 11 libraries covering the recognition sequences of available restriction enzymes. Alongside a set of Python functions to reconstruct the database and more easily access it (which were debugged and had improved documentation added during review). The presented data should be valuable for researchers simply using BACs, as well as those working with larger sections of the genome in terms of synthetic genes, large-scale editing, and mapping.

---

## [Reviewer Report]

Comments on revised manuscriptThe authors have addressed majority of my points. The software installation works great after considering version control. The updated read.me provide detailed information for each function and their required input variables, and the examples in jupyter notebook are a great help for running the code. I did, however, encounter two minor errors when I tested the Ch19_bacmapping_example.ipynb on a Mac system. Please check this and update it.  (1)The .DS_store file that is automatically generated on a Mac system in the bacmapping/Examples/Ch19_example/maps/placed folder causes an error when running bmap.mapPlacedClones(cpustouse=cpus, chunk_size=chunksize). The same problem happened when I ran bmap.mapSequencedClones(cpustouse=cpus). After I deleted .DS_store in the folder, the code worked.  Here is the error message when I ran bmap.mapSequencedClones(cpustouse=cpus). NotADirectoryError: [Errno 20] Not a directory: '/Users/user_nsame/bacmapping/Examples/Ch19_example/maps/sequenced/.DS_Store'  (2) The second error is from running bmap.getRestrictionMap(name,enzyme). I got the error message, 'list' object has no attribute 'item'. I was able to run this function after changing maps[enzyme].item() to maps[enzyme] in line 779 of bacmapping.py. I encountered the same error with the drawMap function. I was able to run to run this function after changing line 847 of bacmapping.py from rmap = maps[nenzyme].item() to rmap = maps[nenzyme].item().  Here is the error message --------------------------------------------------------------------------- AttributeError Traceback (most recent call last) Cell In[20], line 5 3 maps = bmap.getMaps(name) 4 #print(maps) #this is a big dataframe of all the maps, uncomment to check it out -- 5 rmap = bmap.getRestrictionMap(name,enzyme) 6 print('Sites in ' + name + ' where ' + enzyme + ' cuts: '+ str(rmap)) 7 plt = bmap.drawMap(name, enzyme)  File ~/miniconda3/envs/bacmapping/lib/python3.11/site-packages/bacmapping/bacmapping.py:779, in getRestrictionMap(name, enzyme) 777 maps = getMaps(name) 778 nenzyme, r = getRightIsoschizomer(enzyme)  779 return(maps[nenzyme].item())  AttributeError: 'list' object has no attribute 'item'

---

## [Reviewer Report]

Reviewer name and names of any other individual's who aided in reviewer Wei DongDo you understand and agree to our policy of having open and named reviews, and having your review included with the published papers. (If no, please inform the editor that you cannot review this manuscript.)YesIs the language of sufficient quality?YesPlease add additional comments on language quality to clarify if needed
Are all data available and do they match the descriptions in the paper? YesAdditional CommentsAre the data and metadata consistent with relevant minimum information or reporting standards? See GigaDB checklists for examples <a href="http://gigadb.org/site/guide" target="_blank">http://gigadb.org/site/guide</a>YesAdditional CommentsIs the data acquisition clear, complete and methodologically sound?YesAdditional CommentsIs there sufficient detail in the methods and data-processing steps to allow reproduction?YesAdditional CommentsIs there sufficient data validation and statistical analyses of data quality? Not my area of expertiseAdditional CommentsIs the validation suitable for this type of data?YesAdditional CommentsI am not sure about this.This is not my specialty.Is there sufficient information for others to reuse this dataset or integrate it with other data?YesAdditional CommentsAny Additional Overall Comments to the AuthorThis is a great idea, fully exploring, integrating, and utilizing existing data for new research.RecommendationAccept

---

## [Reviewer Report]

Reviewer name and names of any other individual's who aided in reviewer Po-Hsiang HungDo you understand and agree to our policy of having open and named reviews, and having your review included with the published papers. (If no, please inform the editor that you cannot review this manuscript.)YesIs the language of sufficient quality?YesPlease add additional comments on language quality to clarify if needed
Are all data available and do they match the descriptions in the paper? NoAdditional CommentsThe dataset in FTP includes all the Bac sequences and the restriction enzyme recognition sites in csv files. However, I could not find the database of pairs of BACs, which have overlaps generated by restriction enzymes that linearize the BACs. The makePairs function gave me an error when I tried running it locally, so I was not able to verify what is in these datasets. Personally, I find this function to be one of the most useful features described in this manuscript.Are the data and metadata consistent with relevant minimum information or reporting standards? See GigaDB checklists for examples <a href="http://gigadb.org/site/guide" target="_blank">http://gigadb.org/site/guide</a>YesAdditional CommentsThis manuscript contains the necessary minimal information (Submitting author, Author list, Dataset title, Dataset description, and Funding information)Is the data acquisition clear, complete and methodologically sound?YesAdditional CommentsIs there sufficient detail in the methods and data-processing steps to allow reproduction?NoAdditional CommentsThe authors provide their code in GitHub such that researchers can download the datasets and analyze the sequences locally. However, I felt that the descriptions in the readme.md file is often insufficient to reproduce the data presented in the manuscript, especially for researchers with little to no programming experience. Detailed information includes examples of how to use each function, the input format, and the location of the output folder/files. I also encountered software version issues during the installation of bacmapping. Please re-test the code in a new environment and describe all the versions of each software. For instance, I found Python version 3.11 is incompatible with this package while Python version 3.7 is compatible. Is there sufficient data validation and statistical analyses of data quality? NoAdditional CommentsThe author used the BioRestriction class from Biopython to get the digestion site information. No extra validation is conducted in this manuscript. Due to the errors I encountered in re-running the code (see details in Any Additional Overall Comments to the Author), an independent method for checking several digestion sites in some Bac clones is suggested. The suggested independent method is to do enzyme digestion on some Bac clones or upload some Bac sequences to other software and compare the digestion sites.  In the output files that contain the digestions sites for each enzyme, some of the enzyme digestion sites are either NA or []. What is the difference between the two? If they mean the same thing (no cutting by the enzyme), bugs or other coding errors may cause this inconsistency. Please check the code again and also verify some of them using the independent methods suggested above. Examples of this issue are the files in maps>sequenced>CEPHB. Here I list two enzymes that show different results in each file: 3.csv : Ragl ([]), SchI (NA) 6.csv: EspEI (NA), AccII([]) 13.csv: EcoT22I ([]), Hsp92II (NA) X.csv: PacI ([]), AcIWI (NA)
Is the validation suitable for this type of data?NoAdditional CommentsNo validation in this manuscript. See the answer above.Is there sufficient information for others to reuse this dataset or integrate it with other data?NoAdditional CommentsThe authors provide their code in GitHub for people to reproduce the dataset and also to analyze sequences locally. It is very grateful to have the resource of codes to analyze other data and integrate it into the current dataset. However, refining the code and also providing a better user manual is very helpful for people without a lot of coding experience to use it. Please see the suggestions in "Any Additional Overall Comments to the Author" and "Is there sufficient detail in the methods and data-processing steps to allow reproduction?" sections.Any Additional Overall Comments to the AuthorThe authors make a database with enzyme digestion site information of Bac clones to help people to use the Bac clones for further usage. I think it is useful to have this information and also have the code to do further analysis locally. Thus, I think providing a very detailed user manual (or readme.md) is very important to help people use this dataset. Below I summarized the issues I encountered in running codes and also some suggestions.  Major points: (1) I tested some bacmapping functions, and I discovered that some functions are not working as intended due to typos/bugs - The version of the software is required to help people properly install this package - Refining the code and also providing a better user manual is very helpful for people without a lot of coding experience to use it. The detailed information includes examples of how to use each function, the input format, and the location of the output folder/files. Descriptions for some functions in the readme file are not detailed enough and often do not describe what the input needs to be. For example, getCuts() require ‘row’ as input. But the author never gives a detailed description of what ‘row’ is in the readme file. I had to look in bacmapping.py to understand what ‘row’ is. If a function requires the variable ‘row’, show a few examples of how ‘row’ can be extracted from the proper input file. - mapPlacedClones() requires an input file (‘/home/eamon/BACPlay/longboys.csv’, line 335) that is located in the author’s local computer and is not available through github. - Typo in line 814 in getMap(). Should be: name = cloneLine[‘CloneName’] - Inconsistency in output variable type in getMap() (line 830 and 851). When local == ‘sequenced’, the output variable is a tuple, which causes issues in downstream functions such as getRestrictionMap() (line 869). (2) Add pairs of BACs into the dataset (3) The output file of digestion sites of each enzyme, some of the enzyme digestion sites showed NA or [ ]. Please double-check this and explain the differences (4) Validation of an independent method for the digestion map is suggested   Minor points: (1) Add a title to each column of sequencedStats.csv is useful for understanding the table easier
RecommendationMajor Revision